# An excitation mechanism for discrete chorus elements in the magnetosphere

Peter Bespalov[1,2] and Olga Savina[2]

[1]Institute of Applied Physics RAS, Nizhny Novgorod, Russia
[2]National Research University Higher School of Economics, Nizhny Novgorod, Russia

**Correspondence:** P. Bespalov
(PBespalov@mail.ru)

**Abstract.** A beam pulsed amplifier mechanism responsible for effective amplification of short VLF electromagnetic pulses is proposed. Effective amplification near the magnetic equator outside the plasmasphere is considered. A conditional growth rate of short electromagnetic pulses is calculated. Obtained results can explain some important features of the oblique electromagnetic chorus emissions without hiss-like radiation background.

## 1 Introduction

VLF chorus emissions are very intense electromagnetic plasma waves that are naturally excited near the magnetic equatorial plane outside the plasmasphere (Burtis and Helliwell, 1969; Burton and Holzer, 1974; Tsurutani and Smith, 1974). Impressive experimental results of the chorus emission study were obtained within the framework of the CLUSTER project. These results have been presented in detail in many papers (e.g., Santolik, 2009). It is very important that chorus are a succession of discrete emissions.

For electromagnetic chorus with the wave vectors predominantly along the magnetic field significant theoretical results were obtained. The usual kinetic cyclotron instability (Bespalov and Trakhtengerts, 1986) is sometimes possible for plasma parameters in the excitation region, but this instability typically has not a sufficient growth rate to explain the rate of the chorus emission modification. The theory of chorus excitation based on the so-called backward wave oscillator (BWO) mechanism is now well known (see e.g. Trakhtengerts, 1995; Trakhtengerts et al., 2007; Nunn et al., 2009). The implementation of this mechanism is closely related to the hiss-like radiation background and its dynamics. However, the chorus emissions are often recorded without the hiss-like radiation background. In (Omura et al., 2008; Fu et al., 2014) the generation process of the chorus emissions is analyzed by both in theory and simulation assuming that the initial cyclotron wave growth is driven by the strong temperature anisotropy of energetic electrons. To explain the chorus spectrogram, the authors take into account inhomogeneity of the magnetic field and nonlinear wave decay, or the non monotonic energy spectrum of particles. At present, there are significant achievements in the theoretical study of generation electrostatic chorus with the wave vectors near the resonance cone (see, e.g., (Li et al., 2016a)). There are extensive data on the wave normal angle measurements on board THEMIS (Li et al., 2013) and Van Allen Probe (Li et al., 2016b).

Some problems connected with the theoretical analysis of chorus formation remain unsolved, for example, the excitation mechanism of oblique electromagnetic chorus has not been studied. In this paper, we introduce a possible mechanism of oblique electromagnetic chorus excitation without hiss-like radiation background. This mechanism is related to the effective amplification of short electromagnetic pulses from the noise level even in a stable plasma. The amplification takes place in a suitable frequency band near the magnetic equator.

## 2  Wave-particle interaction under special conditions

We consider small-scale wave processes near the local minimum of the magnetic field, which typically is close to the magnetic equator where the plasma is almost homogeneous. We use a linearized Vlasov equation for the disturbed distribution function of energetic non-relativistic electrons $f_\sim$ in a quasineutral cool background plasma

$$
\frac{\partial f_\sim}{\partial t} + v_z \frac{\partial f_\sim}{\partial z} + \mathbf{v}_\perp \frac{\partial f_\sim}{\partial \mathbf{r}_\perp} -
$$
$$
-\frac{e}{mc}\mathbf{v}\times\mathbf{B}\frac{\partial f_\sim}{\partial \mathbf{v}} - \frac{e}{m}(\mathbf{E}_\sim + \frac{1}{c}\mathbf{v}\times\mathbf{B}_\sim)\frac{\partial f_\circ}{\partial \mathbf{v}} = 0 \,, \tag{1}
$$

where $e$ is the absolute value of the electron charge, $m$ is the electron mass, $c$ is the speed of light in free space, $v_z = dz/dt$ and $\mathbf{v}_\perp = d\mathbf{r}_\perp/dt$ are the components of electron velocity $\mathbf{v}$ along and across to the ambient magnetic field $\mathbf{B}$, $f_\circ(\mathbf{v})$ is the undisturbed distribution function, $\mathbf{E}_\sim$ and $\mathbf{B}_\sim$ are the electric and magnetic field disturbances.

Let a short VLF electromagnetic pulse propagate in a homogeneous plasma along the $z$ axe directed along the magnetic field. We assume that the short electromagnetic pulse has an envelope determined by the function $A(\xi)$ with a nearly step-like form, finite duration and a unit value in the main body of the pulse

$$
\mathbf{E}_\sim = A(\xi)\mathbf{E}_{[\sim]}(t,\mathbf{r}) \,, \quad \mathbf{B}_\sim = A(\xi)\mathbf{B}_{[\sim]}(t,\mathbf{r}) \,, \tag{2}
$$

where $\xi = z - v_{gz}t$ , $v_{gz}$ is the component of the group velocity along the magnetic field. The chosen pulse shape is modeled and corresponds to the shot noise well known in electronics (see, e.g., (Rytov et al., 1989)). It is convenient to replace $f_\sim$ in Eq. (1) by an expression similar to Eq. (2):

$$
f_\sim = A(\xi)f_{[\sim]}(t,\mathbf{r},\mathbf{v}) \,. \tag{3}
$$

Upon substitution of Eqs. (2) and (3) in Eq. (1) we obtain

$$
\frac{\partial A}{\partial \xi}(-v_{gz} + v_z)f_{[\sim]} + A\{\frac{\partial f_{[\sim]}}{\partial t} + v_z\frac{\partial f_{[\sim]}}{\partial z} + \mathbf{v}_\perp\frac{\partial f_{[\sim]}}{\partial \mathbf{r}_\perp} -
$$
$$
-\frac{e}{mc}\mathbf{v}\times\mathbf{B}\frac{\partial f_{[\sim]}}{\partial \mathbf{v}} - \frac{e}{m}(\mathbf{E}_{[\sim]} + \frac{1}{c}\mathbf{v}\times\mathbf{B}_{[\sim]})\frac{\partial f_\circ}{\partial \mathbf{v}}\} = 0 \,, \tag{4}
$$

We restrict ourselves to the considering the energetic electrons for which

$$v_z \simeq v_{gz} \, . \tag{5}$$

For these particles, the first term in Eq. (4) is absent because the disturbed distribution function is nonzero only in a small vicinity of the velocity which satisfies to the condition (5). So, under condition (5) the interaction of a short electromagnetic pulse with energetic electrons is described by Vlasov equation (4) without the first term. It is very important that the reduced equation (4) (without the first term) is identical to Vlasov equation for wave-particle interaction in a homogeneous plasma.

Note that the particles which do not satisfy to Eq. (5) exist in the plasma, but their interaction with a short electromagnetic pulse is often not important, and we do not take this interaction into account.

Now we point the condition favorable for the existence of a short electromagnetic pulse that propagates along the magnetic field at a constant velocity without the additional phase modulation and smallest dispersion distortion of the pulse front. The solution of this problem is known. According to electrodynamics (see, e.g., (Jacson, 1962; Sommerfeld, 1914)) this takes place

under condition $v_{ph_z} = v_{gz}$. This relationship agrees with Eq. (5), if the following two equalities are valid simultaneously:

$$v_{ph_z} = v_{gz} = v_z \, , \tag{6}$$

where $v_{ph_z}$ is the component of the phase velocity along the magnetic field. The particles interact with short electromagnetic pulse under condition (6) as in a homogeneous plasma.

If the ion motion in a relatively dense background plasma is not important, then the dispersion equation of electromagnetic waves for the frequency band $\omega_{LHF} < \omega < \omega_B$ (where $\omega_{LHF}$ is the lower-hybrid frequency and $\omega_B$ is the absolute value of the

electron cyclotron frequency) in the quasi-longitudinal approximation (Ginzburg, 1970; Helliwell, 1965) takes the well-known form

$$\omega = \omega_w(k_z, k_\perp) \equiv \frac{\omega_B \, |k_z| \left( k_z^2 + k_\perp^2 \right)^{1/2}}{k_z^2 + k_\perp^2 + (\omega_p/c)^2} \, . \tag{7}$$

Here, $k_z$ and $k_\perp$ are the whistler wave vector components along and across the magnetic field, respectively, and $\omega_p$ is the electron plasma frequency at the magnetic equator.

Note the first key point of our analysis. It is known (Helliwell, 1995) that according to the dispersion equation (7) the

conditions (6) are fulfilled independent of $k_\perp$ for two selected velocities along the magnetic field

$$\frac{\omega}{k_z} = \frac{\partial \omega}{\partial k_z} = v_z = \pm u_G \, , \quad u_G = \frac{c\omega_B}{2\omega_p} \, , \tag{8}$$

where $u_G$ is the Gendrin velocity. We will take into account only one positive velocity in the intermediate calculations.

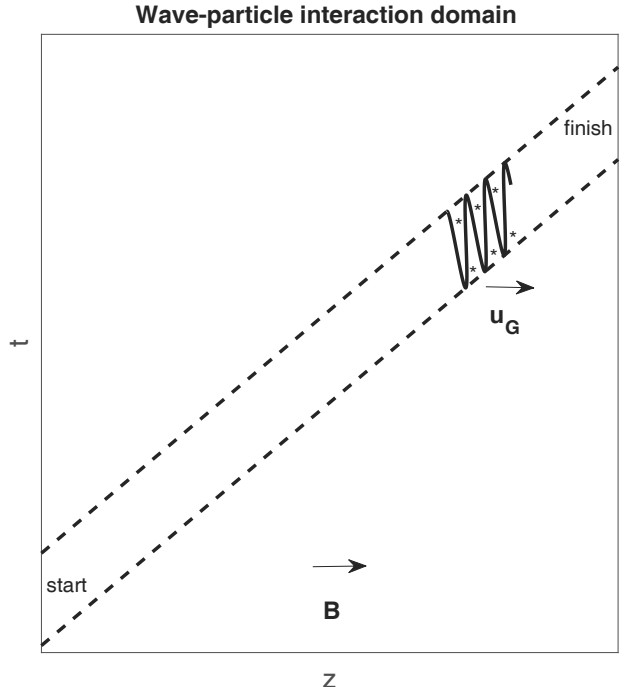

**Figure 1.** The resonance beam of energetic electrons (stars) and the short electromagnetic pulse (sine) move together in the domain along the corridor between the start and finish lines.

## 3   The beam pulsed amplifier mechanism of the oblique electromagnetic chorus excitation due to the conditional instability of a short electromagnetic pulse

On the plane $z$, $t$, the specific features of particles and wave, which satisfy Eqs. (8), are explained in Fig. 1. Here $z$ is the axis along the magnetic field, the start and finish lines correspond to the wave-particle interaction region boundaries, $t$ is the time
5   counted from the pulse crossing the start line. The resonance particles (3) and the short electromagnetic pulse (2) are situated in the domain which moves along a narrow corridor between two parallel dashed lines. The corridor width is determined by the pulse duration. These particles and waves in the domain form a separated plasma subsystem.

Effective wave–particle interaction in a homogeneous plasma in the magnetic field takes place at the resonance conditions $\omega - k_z v_z = s\omega_B$, where $s$ is the integer. Such a resonance condition is part of the mentioned equalities (8) for the $\check{C}$erenkov
10   resonance $s = 0$. We limit ourselves to this possibility only:

$$\omega - k_z v_z = 0 \,. \tag{9}$$

We expect that the following two simplifications are fulfilled: the radiation power of an individual energetic electron corresponds to the so-called dipole approximation; the specific kinetic effects like the Landau damping are not important. Both of the mentioned simplifications take place for energetic electrons with a sufficiently small dispersion over the transverse and longitudinal velocities (Ginzburg, 1970):

$$k_\perp^2 \overline{v_\perp^2} < \omega_B^2 \,, \quad k_z^2 \overline{(v_z - u_G)^2} < \gamma^2 \,, \tag{10}$$

where the overbar means averaging and $\gamma$ is the instability growth rate. The first inequality makes it possible to expand the Bessel functions for small arguments (this so-called dipole approximation is used for obtaining the expressions of the permittivity tensor). The second inequality makes it possible not to take into account the kinetic effects (like the Landau damping) for instability with a large hydrodynamic growth rate.

Note the second key point for a qualitative calculation of short electromagnetic pulse amplification. According to the previous
comments, the electromagnetic field with suitable frequency in the short pulse changes its value as the electromagnetic wave field in a homogeneous plasma with electron beam described by the effective distribution function:

$$f_b = \frac{n_b}{2\pi v_\perp} \delta(v_\perp) \delta(v_z - u_G) \,, \tag{11}$$

where $n_b$ is the density of the resonance electron beam. It can be shown that for the wave-particle interaction at the Čerenkov resonance, the results of calculations do not depend on the actual form of the distribution function over perpendicular velocities if the first unequality (10) takes place. We select the effective distribution function (11) with delta function over perpendicular
velocities to simplify the further algebra. On the other hand, the delta function over velocities along the magnetic field exactly follows from the condition (8) for the selected velocity.

We explain additionally the expression for the effective distribution function (11). Assume that a short non-spreading pulse propagates at a constant velocity ($v_{phz} = v_{gz} = u_G$) along the magnetic field in a plasma with an arbitrary undisturbed distribution function $f_\circ(\mathbf{v})$. The wave-particle interaction within a short pulse is determined by the characteristic time of the electron
velocity variation $\tau$. For correct accounting of the wave-particle interaction, it is necessary to know the distribution function averaged over the time scale $\tau$. Let us consider the effective distribution function inside the pulse. Inside a current pulse localization there is a beam of electrons which have entered the interaction region together with the pulse and move jointly with it. The contribution of these electrons to the effective distribution function is proportional to $n_b \delta(v_z - u_G)$. The suprathermal electrons with other velocities or flight moments do not give an appreciable contribution to the effective distribution function
within the pulse since they cross the pulse in a narrow corridor in Fig. 1 too quickly on a time scale $\tau$ or do not have contact with the pulse. Thus, we take the effective distribution function (11) to analyze the evolution of a short electromagnetic pulse.

We expect that the quasineutral plasma consists of three fractions: unmoved protons; cool electrons; weak electron beam along the magnetic field without thermal dispersion, described by the effective distribution function (11). So, the complicated

geophysical problem is reduced to a typical problem of plasma physics (e.g., Akhiezer et al., 1975). We now consider the permittivity tensor $\varepsilon_{\alpha\beta}$ of such a medium taking into account that the resonance beam density is relatively small. The beam-related terms in the permittivity tensor produce any effect only if they have a resonance pole at $\omega - k_z u_G = 0$ (see Eq. (9)). There is one such resonance term only in the component $\varepsilon_{33}$. The permittivity tensor with mentioned resonance term has the following non-zero components:

$$\varepsilon_{11} = \varepsilon_{22} = 1 - \frac{\omega_p^2}{\omega^2 - \omega_B^2} \,,$$

$$\varepsilon_{12} = -\varepsilon_{21} = i\frac{\omega_B \omega_p^2}{\omega(\omega^2 - \omega_B^2)} \,, \tag{12}$$

$$\varepsilon_{33} = 1 - \frac{\omega_p^2}{\omega^2} - \frac{n_b \omega_p^2}{n_p(\omega - k_z u_G)^2} \,,$$

where $n_p$ is the background plasma density. The plasma dispersion equation (Ginzburg, 1970) for a plane electromagnetic wave in which all the variables are proportional to $\exp\{i(-\omega t + k_\perp x + k_z z)\}$, takes the well-known form:

$$\det \left\| k^2 \delta_{\alpha\beta} - k_\alpha k_\beta - (\omega/c)^2 \varepsilon_{\alpha\beta} \right\| = 0 \,. \tag{13}$$

This determinant in the quasi-longitudinal approximation is reduced to the following dispersion equation $(\omega - \omega_w(k_z, \theta))(\omega - k_z u_G)^2 = (n_b/4n_p)\omega^3 \sin^2\theta$ and finally, in accordions with Eqs. (7) and (8), we have

$$(\omega - \omega_w(k_z, \theta))(\omega - |k_z| u_G)^2 = \frac{n_b \omega_B^3}{32 n_p} \sin^2\theta |\cos\theta|^3 \,, \tag{14}$$

where the absolute value symbols take into account two selected velocities (8) along the magnetic field. Recall that the dispersion equation (7) was found in the same quasi-longitudinal approximation. Exactly in accordions with Eq. (7) we have

$$\omega_w(k_z, \theta) = \frac{\omega_B (k_z c/\omega_p)^2 |\cos\theta|}{(k_z c/\omega_p)^2 + \cos^2\theta} \,, \tag{7a}$$

where $\cos\theta = k_z/(k_z^2 + k_\perp^2)^{1/2}$, $\theta$ is the angle between the wave vector and the magnetic field.

The dispersion equation (14) is a cubic equation for the frequency $\omega$. This equation has three roots, one of which (see Fig. 2) for conditions close to optimal (8) corresponds to the short electromagnetic pulse instability with the conditional growth rate

$$\gamma = \frac{\sqrt{3}}{4}\left(\frac{n_b}{4n_p}\sin^2\theta |\cos\theta|^3\right)^{1/3}\omega_B \,. \tag{15}$$

According to Eq. (15), the conditional growth rate of the short electromagnetic pulse instability is maximum for $\cos^2\theta_{BPA} = 0.6$. So, according to Eqs. (8) and (15), it is easy to find the following optimal wave characteristics favorable to the maximum conditional growth rate of a short electromagnetic pulse

$$\theta_{BPA} \simeq \begin{cases} 39° \,; & k_{zBPA} \simeq \pm 0.8\omega_p/c \,; \\ 141° \,; & k_{\perp BPA} \simeq 0.6\omega_p/c \,; \end{cases}$$

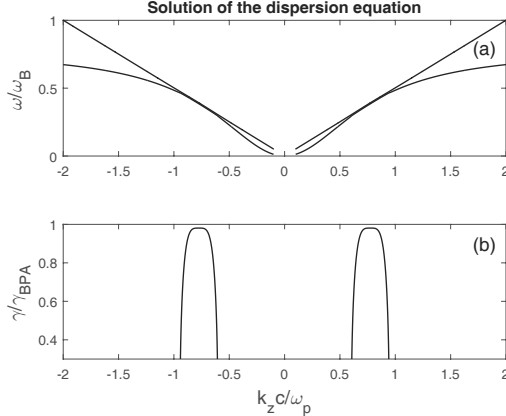

**Figure 2.** Numerical solution of the dispersion equation (14) for $n_b/n_p = 10^{-5}$ and $\theta = \theta_{BPA}$: (a) The frequencies (real part of frequencies) dispersion behavior of the longitudinal wave number; (b) Conditional growth rate of a short electrostatic pulse as a function of the longitudinal wave number.

$$\omega_{BPA} \simeq 0.39\omega_B \ , \ \gamma_{BPA} \simeq 0.15\omega_B(n_b/n_p)^{1/3} \ . \tag{16}$$

Note, that the resonance beam density $n_b$ is significantly less than the density of energetic electrons but according to the additional calculations, it is enough for explanation of the experimentally recored values of $\gamma \sim 10^2 \ \mathrm{s}^{-1}$ in the chorus excitation region. Now it is possible to estimate an electromagnetic pulse duration $t_p$ . The beam pulsed amplifier mechanism is effective for electromagnetic pulses with the duration $t_p \leq 10/\gamma_{BPA} \sim 0.1$ s at the linear stage of the pulse evolution.

## 4   Conclusions and discussion

The proposed beam pulsed amplifier (BPA) mechanism produces effective amplification of short electromagnetic pulses with frequency close to $\omega_{BPA}$ in the VLF frequency band near the magnetic equator. The key conditions (8), (16) take place outside the plasmasphere, where $u_G$ is close to the typical velocity of energetic non-relativistic electrons. In this region the ratio $(n_b/n_p)^{1/3}$ is not so small. This leads to the amplification of short electromagnetic pulses from the noise level and the formation of discrete VLF emissions.

The threshold of BPA mechanism is mainly determined by the kinetic Čerenkov damping ($s = 0$) of electromagnetic waves in the epithermal background plasma. The magnitude of the decay rate is considerably less than the growth rate $\gamma_{BPA}$ in the chorus excitation region.

By the further analysis it is possible to explain the gap between the lower and upper chorus frequency bands which are well known experimentally (e.g., (Bell et al., 2009). Actually, the short electromagnetic pulse excitation takes place for frequency

bands both below and above $\omega_{BPA}$ because of the spectral pulse distortion and the resonance beam velocity decrease as the pulse moves through the wave-particle interaction region.

Note that electromagnetic signals with a smooth envelope are not amplified due to BPA mechanism during their propagation through the region near the magnetic equator. Therefore, it is possible to explain the excitation of chorus emissions without a hiss-like background.

BPA mechanism is not connected directly with the energetic electron precipitation into the ionosphere because this mechanism is responsible for the distribution function modification in the velocity space only far enough from the loss cone. On the other hand, exactly after the amplification, a strong electromagnetic pulse during its propagation to the ionosphere can interact with more energetic electrons due to the cyclotron resonance ($s = 1$). Such an interaction produces precipitation of energetic electrons into the ionosphere that is conjugate to the electromagnetic pulse.

*Acknowledgements.* P.A. Bespalov was supported by RSF, project no. 16-12-10528 (Sect. 2), the Russian Ministry of Science and Education, project no. 14.Z50.31.0007 (Sect. 3), and Fundamental Research Program no. 28 (numerical calculations).

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
