# Peer review of "An excitation mechanism for discrete chorus elements in the magnetosphere"

_Annales Geophysicae, 2018_

## Short Comment (SC1) · 22 Jun 2018

This short paper proposes an interesting new mechanism for excitation of whistler-mode pulses outside of the plasmasphere where there is no hiss background from which such signals may grow. Instead, it specifically proposes a mechanism for excitation of short pulses (like chorus elements) under special wave conditions due to an energetic electron beam. This seems to be an interesting new possibility that merits attention via the literature.

---

## Referee Comment (RC1) · Anonymous Referee #1 · 10 Jul 2018

The manuscript claims to have proposed an VLF electromagnetic pulse excitation mechanism in the magnetosphere; however, the manuscript lacks data supporting its idea, and misses many key references for its claims and assumptions. I could only find an exhibition of a wave instability growth rate calculation. There is neither simulation based on this calculated growth rate nor any observation to be compared with, therefore it is hard to guess any physical validity of this calculation, yet the authors still claim that it could explain many phenomena in their conclusions. The calculation relies on many key simplifications and assumptions that lack enough reference to support, for example in the paragraph around Eq. (10). The calculation procedure appears to me as a composition of selected equations and results from textbooks, without enough

[Figure]

explanation of their backgrounds or even physical meaning of symbols. Moreover, I don't see a clear logical link between these equations, for example, the Vlasov equation discussed in Eq. (1)∼(4) are never used in the later part of the manuscript, and it is hard to understand why they are there. In its current form, it is difficult to fairly evaluate the originality and scientific contribution/significance of the manuscript. Therefore I suggest the authors to resubmit after serious and substantial improvement. Additional comments are shown below.

Figure 1, which is the only figure in the manuscript, is not clear. Why particles (marked stars) move as shown in Figure 1? What are the two parallel lines?

Why a nearly step-like form of A in equation (2) is important in the beam pulsed amplifier mechanism proposed? This form seems lack of observational support.

The instability requires electrons, or even ideal beam with vz=Ug, which is not realistic either. Another mainstream idea for chorus generation, which is associated with inhomogeneity and cyclotron instability, is not discussed.

As equation 16 shows, the favorable wave normal from the proposed mechanism is quite oblique, which is not consistent with often observed quasiparallel propagation in the literature.

---

## Author Comment (AC1) · 31 Jul 2018

**Respond to all referee comments**

**(1) Comment from Referee**

The manuscript claims to have proposed an VLF electromagnetic pulse excitation mechanism in the magnetosphere; however, the manuscript lacks data supporting its idea, and misses many key references for its claims and assumptions. I could only iňĄnd an exhibition of a wave instability growth rate calculation. There is neither simulation based on this calculated growth rate nor any observation to be compared with,

therefore it is hard to guess any physical validity of this calculation, yet the authors still claim that it could explain many phenomena in their conclusions. The calculation relies on many key simpliñAcations and assumptions that lack enough reference to support, for example in the paragraph around Eq. (10). The calculation procedure appears to me as a composition of selected equations and results from textbooks, without enough explanation of their backgrounds or even physical meaning of symbols. Moreover, I don't see a clear logical link between these equations, for example, the Vlasov equation discussed in Eq. (1) $\hat{a}$ Lij(4) are never used in the later part of the manuscript, and it is hard to understand why they are there. In its current form, it is difiñAcult to fairly evaluate the originality and scientiñAc contribution/signiñAcance of the manuscript. Therefore I suggest the authors to resubmit after serious and substantial improvement. Additional comments are shown below.

Figure 1, which is the only in Agure in the manuscript, is not clear. Why particles (marked stars) move as shown in Figure 1? What are the two parallel lines?

Why a nearly step-like form of A in equation (2) is important in the beam pulsed ampliiňĄer mechanism proposed? This form seems lack of observational support.

The instability requires electrons, or even ideal beam with vz=Ug, which is not realistic either. Another mainstream idea for chorus generation, which is associated with inhomogeneity and cyclotron instability, is not discussed.

As equation 16 shows, the favorable wave normal from the proposed mechanism is quite oblique, which is not consistent with often observed quasiparallel propagation in the literature.

**(2) Author's response**

We would like to thank the Reviewer for the time he/she spent reading, and commenting our manuscript. We have prepared a point-by-point answer to his/her comments below. The responses are marked in bold. Reviewer's Comments:

The manuscript claims to have proposed an VLF electromagnetic pulse excitation mechanism in the magnetosphere; however, the manuscript lacks data supporting its idea, and misses many key references for its claims and assumptions.

**Response:**

This short theoretical manuscript is addressed to professionals who are well aware of the published achievements of experimental and theoretical research on the chorus radiation in the magnetosphere. Therefore, the bibliography contained only references that are important for the original study. Nevertheless, we have extended the bibliography to make it more balanced in term of presenting the results of various scientific schools.

Reviewer's Comments:

I could only find an exhibition of a wave instability growth rate calculation.

**Response:**

The calculation of the conditional instability growth rate is the main new achievement of this work.

Reviewer's Comments:

There is neither simulation based on this calculated growth rate nor any observation to be compared with, therefore it is hard to guess any physical validity of this calculation, yet the authors still claim that it could explain many phenomena in their conclusions.

**Response:**

The study of the chorus emissions is the important geophysical problem. Several mechanisms of chorus excitation are discussed in the literature. Each of them explains some characteristics of the phenomenon, but has its limitations

СЗ

on the conditions of applicability and the level of consistency with the experimental data. This manuscript is not a overview and we do not comment on known mechanisms, but draw attention to a new reason of chorus excitation.

In our opinion, before carrying out simulations, it is useful to understand the nature of chorus excitation. In the manuscript we try to do this. A comprehensive study of the simulation is beyond the scope of the present paper and it is left for future work. In turn, a detailed comparison of the theoretical conclusions with the experimental data is planned in another article.

Reviewer's Comments:

The calculation relies on many key simplifications and assumptions that lack enough reference to support, for example in the paragraph around Eq. (10).

**Response:**

Two inequalities (10) are the classical positions of plasma physics. The first inequality (10) makes it possible to expand the Bessel functions for small arguments (this so-called dipole approximation is used for obtaining the expressions of the permittivity tensor). The second inequality (10) makes it possible not to take into account the kinetic effects (like the Landau damping) for instability with a large hydrodynamic growth rate. We added this explanation to the text.

**Reviewer's Comments:**

The calculation procedure appears to me as a composition of selected equations and results from textbooks, without enough explanation of their backgrounds or even physical meaning of symbols.

**Response:**

We additionally checked the notations and made small corrections.

Reviewer's Comments:

Moreover, I don't see a clear logical link between these equations, for example, the Vlasov equation discussed in Eq. (1)-(4) are never used in the later part of the manuscript, and it is hard to understand why they are there.

**Response:**

As a result of the analysis of the Eqs. (1)-(4), we showed that the spectral components of the wave packets for which the Eqs. (8) are fulfilled interact with particles as in a homogeneous plasma with the distribution function (11). Thus, we reduced the problem of the electromagnetic pulse amplification to the standard task of plasma stability.

Reviewer's Comments:

In its current form, it is difficult to fairly evaluate the originality and scientific contribution/significance of the manuscript.

**Response:**

It is clear that the perception of a new mechanism of oblique electromagnetic chorus excitation is not easy, but a reasonable solution is required by researchers.

Reviewer's Comments:

Therefore I suggest the authors to resubmit after serious and substantial improvement.

Additional comments are shown below. Figure 1, which is the only figure in the manuscript, is not clear. Why particles (marked stars) move as shown in Figure 1? What are the two parallel lines?

Response:

Note that there is also Fig. 2 in the manuscript.

On the plane z, t the specific features of particles and wave, which satisfy Eqs.

(8), are explained in Fig. 1. Here z is the axis along the magnetic field, the start and finish lines correspond to the wave-particle interaction region boundaries, tis the time is counted from the pulse crossing the start line. The resonance particles (3) and the short electromagnetic pulse (2) are situated in the domain which moves along a narrow corridor between two parallel dashed lines. The corridor width is determined by the pulse duration. These particles and waves in the domain form a separated plasma subsystem. We added text to the manuscript.

Reviewer's Comments:

Why a nearly step-like form of A in equation (2) is important in the beam pulsed amplifier mechanism proposed? This form seems lack of observational support.

**Response:**

The chosen pulse shape is modeled and corresponds to the shot noise well known in electronics (see, e.g., (Rytov et al., 1989)). We added text to the manuscript. We extended the bibliography.

Reviewer's Comments:

The instability requires electrons, or even ideal beam with vz=Ug, which is not realistic either. Another mainstream idea for chorus generation, which is associated with inhomogeneity and cyclotron instability, is not discussed.

**Response:**

The beam under consideration is effective. It is effective only for waves from short electromagnetic pulses with appropriate frequencies and angles of the wave normal. At the same time, if an electromagnetic disturbance with other properties falls on the wave-particle interaction region, then the beam is not effective and the disturbance is decayed.

The study of the chorus emissions is the important geophysical problem. Several mechanisms of chorus excitation are discussed in the literature. Each of them explains some characteristics of the phenomenon, but has its limitations on the conditions of applicability and the level of consistency with the experimental data. This manuscript is not a overview and we do not comment on known mechanisms (see, e.g., (Omura et al., 2008; Fu et al., 2014)), but draw attention to a new reason of chorus excitation. We have extended the bibliography.

Reviewer's Comments:

As equation 16 shows, the favorable wave normal from the proposed mechanism is quite oblique, which is not consistent with often observed quasiparallel propagation in the literature.

**Response:**

There are extensive data on the wave normal angle measurements on board THEMIS and Van Allen Probe. There is no theoretical model for oblique electromagnetic chorus. We replaced "chorus" to "oblique electromagnetic chorus" in several places.

(3) Author's changes in manuscript

Page 1 line 3

The text:

"chorus"

is replaced by

"oblique electromagnetic chorus"

Page 1 line 11

The text:

"Some problems connected with the theoretical analysis of chorus excitation remain

unsolved."

is removed

Page 1 line 11

The text:

"For electromagnetic chorus with the wave vectors predominantly along the magnetic field important theoretical results were obtained."

is added

Page 1 line 17

The text:

"At present, there are significant achievements in the theoretical study of generation electrostatic chorus with the wave vectors near the resonance cone (see, e.g., (Li et al., 2016))."

is added

Page 1 line 17

The text:

"In (Omura et al., 2008; Fu et al., 2014) the generation process of the chorus emissions is analyzed by both in theory and simulation assuming that the initial cyclotron wave growth is driven by the strong temperature anisotropy of energetic electrons. To explain the chorus spectrogram, the authors take into account inhomogeneity of the magnetic field and nonlinear wave decay, or the non monotonic energy spectrum of particles."

is added:

Page 1 line 18

The text:

"Some problems connected with the theoretical analysis of chorus excitation remain unsolved, for example, the excitation mechanism of oblique electromagnetic chorus has not been studied. "

is added

Page 1 line 18

The text:

"chorus"

is replaced by

"oblique electromagnetic chorus"

Page 2 line 5

The text:

"are the electron velocity"

is replaced by

"are the components of the electron velocity v"

Page 2 line 7

The text is added:

"E and B are the electric and magnetic field disturbances"

is added

Page 2 lines 8-12

The expression:

 $z - v_{gz}t$

is replaced by " $\xi$ " Page 2 line 10 The text:

"The chosen pulse shape is modeled and corresponds to the shot noise well known in electronics (see, e.g., (Rytov et al., 1989))."

is addeed

Page 3 line 18

The text:

"Here z is the axis along the magnetic field, the start and finish lines correspond to the wave-particle interaction region boundaries, t is the time counted from the pulse crossing the start line." is addeed

Page 3 line 20

The text: "The corridor width is determined by the pulse duration."

is addeed

Page 4 line 5

The text:

"The first inequality makes it possible to expand the Bessel functions for small arguments (this so-called dipole approximation is used for obtaining the expressions of the permittivity tensor). The second inequality makes it possible not to take into account the kinetic effects (like the Landau damping) for instability with a large hydrodynamic growth rate. "

is added

Page 8 line 12

The text:

"Fu, X., Cowee, M.M., Friedel, R.H., Funsten, H.O., Gary, S.P., Hospodarsky, G.B., Kletzing, C., Kurth, W., Larsen, B.A., Liu, K., MacDonald, E.A., Min, K., Reeves, G.D., Skoug, R.M., and Winske, D. (2014). Whistler anisotropy instabilities as the source of banded chorus: Van Allen Probes observations and Particle-in-Cell simulations. Journal of Geophysical Research, 119, 8288-8298, doi: 10.1002/2014JA020364."

is added

Page 8 line 17

The text:

"Li, W., Mourenas, D., Artemyev, A.V., Bortnik, J., Thorne, R.M., Kletzing, C.A., Kurth, W.S., Hospodarsky, G.B., Reeves, G.D., Funsten, H.O., and Spence, H.E.: Unraveling the excitation mechanisms of highly oblique lower band chorus waves, Geophys. Res. Lett., 43, 8867. doi: 10.1002/2016GL070386, 2016."

is added

Page 8 line 19

The text:

"Omura, Y., Katoh, Y., and Summers D. (2008). Theory and simulation of the generation of whistler-mode chorus. Journal of Geophysical Research, 113, A04223, doi:10.1029/2007JA012622."

is added

Some minor corrections were made to edit the manuscript.

Please also note the supplement to this comment:

https://www.ann-geophys-discuss.net/angeo-2018-38/angeo-2018-38-AC1-supplement.zip

---

## Author Response (AR1)

**Respond to editor comments**

(1) Comment from Editor

I have gone through your responses to the referee and the modifications to your manuscript and the issue of how realistic the proposed mechanism would be for excitation of chorus waves in the environment of the inner magnetosphere still remains.

For example:

* Your response to the referee's comment "The instability requires electrons, or even ideal beam with vz=Ug, which is not realistic either" was that "The beam under consideration is effective…". However, I don't think the point of the referee was whether the beam is effective or not but rather, whether such beams can actually be observed in the inner magnetosphere. If they are not observed, then the mechanism is not realistic for the inner magnetosphere environment.

* Your response to the referee's comment "the favorable wave normal from the proposed mechanism is quite oblique, which is not consistent with often observed quasi-parallel propagation in the literature" was "There are extensive data on the wave normal angle measurements on board THEMIS and Van Allen Probes. There is no theoretical model for oblique electromagnetic chorus." Nonetheless, no references of such observations are provided.

(2) Author's response

**We would like to thank the Editor for important comments. We have prepared a point-by-point answer to comments below. The responses are marked in bold.**

Editor's Comments:

I have gone through your responses to the referee and the modifications to your manuscript and the issue of how realistic the proposed mechanism would be for excitation of chorus waves in the environment of the inner magnetosphere still remains.

**Response:**
**Accumulated experimental data and theoretical experience showed that progress in the chorus excitation problem solution is due to some modification of the formalism.**

Editor's Comments:

Your response to the referee's comment "The instability requires electrons, or even ideal beam with vz=Ug, which is not realistic either" was that "The beam under consideration is effective…". However, I don't think the point of the referee was whether the beam is effective or not but rather, whether such beams can actually be observed in the inner magnetosphere. If they are not observed, then the mechanism is not realistic for the inner magnetosphere environment.

**Response:**
**We explain additionally the expression for the effective distribution function (11). Assume that a short non-spreading pulse propagates at a constant velocity ($v_{phz} = v_{gz} = u_G$) along the magnetic field in a plasma with an arbitrary undisturbed distribution function $f_o(\vec{v})$. The wave-particle interaction within a short pulse is determined by the characteristic time of the electron velocity variation $\tau$. For correct accounting of the wave-particle interaction, it is necessary to know the distribution function averaged over the time scale $\tau$. Let us consider the effective distribution function inside the pulse. Inside the pulse there is a population of electrons which have flown into the interaction region together with the pulse and is fly together with it. The contribution of these particles to the effective distribution function is proportional to $n_b \delta(v_z - u_G)$. The suprathermal electrons with other velocities or flight moments do not give an appreciable contribution to the effective distribution function within the pulse since they cross the pulse in a narrow corridor in Fig. 1 too quickly on a time scale $\tau$ or do not have contact with the pulse. Thus, we take the effective distribution function (11) to analyze the evolution of a short electromagnetic pulse. We added this text to the manuscript.**

Editor's Comments:

Your response to the referee's comment "the favorable wave normal from the proposed mechanism is quite oblique, which is not consistent with often observed quasi-parallel propagation in the literature" was "There are extensive data on the wave normal angle

measurements on board THEMIS and Van Allen Probes. There is no theoretical model for oblique electromagnetic chorus." Nonetheless, no references of such observations are provided.

**Response:**
**We added references to the manuscript and extended the bibliography.**

 (3) Author's changes in manuscript

Page 1 line 21
The text
"There are extensive data on the wave normal angle measurements on board THEMIS (Li et al., 2013) and Van Allen Probe (Li et al., 2016b)."
is added

Page 5 line 15
The text
"We explain additionally the expression for the effective distribution function (11). Assume that a short non-spreading pulse propagates at a constant velocity ( $v_{phz} = v_{gz} = u_G$ ) along the

magnetic field in a plasma with an arbitrary undisturbed distribution function $f_{\circ}(\vec{v})$. The wave-particle interaction within a short pulse is determined by the characteristic time of the electron velocity variation $\tau$. For correct accounting of the wave-particle interaction, it is necessary to know the distribution function averaged over the time scale $\tau$. Let us consider the effective distribution function inside the pulse. Inside the pulse there is a population of electrons which have flown into the interaction region together with the pulse and is fly together with it. The contribution of these particles to the effective distribution function is proportional to $n_b \delta(v_z - u_G)$. The suprathermal electrons with other velocities or flight moments do not give an appreciable contribution to the effective distribution function within the pulse since they cross the pulse in a narrow corridor in Fig. 1. too quickly on a time scale $\tau$ or do not have contact with the pulse. Thus, we take the effective distribution function (11) to analyze the evolution of a short electromagnetic pulse."
is added

Page 8 line 20
The text

"Li, W., Bortnik, J., Thorne, R.M., Cully, C.M., Chen, L., Angelopoulos, V., Nishimura, Y., Tao, J.B., Bonnell, J.W. and LeContel, O.: Characteristics of the Poynting flux and wave normal vectors of whistler-mode waves observed on THEMIS, J. Geophys. Res., 118, 1461, doi:10.1002/jgra.50176, 2013."
is added

Page 8 line 24
The text
"Li, W., Santolik, O., Bortnik, J., Thorne, R.M., Kletzing, C.A., Kurth, W.S., and Hospodarsky, G.B.: New chorus wave properties near the equator from Van Allen Probes wave observations. Geophys. Res. Lett., 43, 4725, doi: 10.1002/2016GL0687, 2016."
is added

Some minor corrections were made to edit the manuscript.

**Respond to referee#2 comments**

(1) Comment from Referee#2

A mechanism responsible for amplification of short VLF electromagnetic pulses based on the cold beam is proposed. A conditional growth rate of short electromagnetic pulses of whistler wave propagating with the wave normal angle equal to the Gendrin angle (presenting the less dispersive wave packet) is calculated. This condition is often discussed as the mechanism of the creating the wave power gap around 0.5 fce. Using the assumption of the cold beam (actually delta function) and making this the key point in the approximation makes the results almost inapplicable to the inner magnetosphere processes. A couple of questions should be addressed before the manuscript might be considered for publication (please, see detailed comments below). Some inaccuracy in the terminology used in the text makes hard to read the manuscript.

The introduction is pure. I think some description of recent results presenting the chorus wave properties, the chorus generation mechanisms (especially beam driven) have to be presented and discussed. Some discussion of the previous results based on the similar initial conditions (for example, Starodubtsev, et al. (1999) Resonant Cherenkov emission of whistlers by a modulated electron beam. Physics of Plasmas, 6(7), 2862–2869. https://doi.org/10.1063/1.873244 and Pivovarov et al., (2012). Excitation of VLF waves by an electron beam injected into the ionosphere. Journal of Geophysical Research: Space Physics, 100(A9), 17515–7526.https://doi.org/10.1029/95JA01156 and some others from the references therein) have to be included and the new effects obtained in the manuscript need to be emphasized.

Condition (5) in the context looks too artificial and needs some more discussion because so cold electron beams are not observed in the magnetosphere that makes all results to be related more to laboratory plasma.

The condition of 'a short electromagnetic pulse' is not well established and explained

The term 'the Doppler resonances" is not often used for $w - k_z v_z = swB$. I suggest to change that to the more conventional ones.

L45: "Cos Theta _BPA=0.6" – please, specify is that definition for the angle of the maximal growth rate? This is not clear here and after. Actually, the authors discuss a very specific case of propagation at the Gendrin angle. This condition gives only one solution (if any) for the particular beam velocity, i.e. properties of the amplified wave are determined by the beam parameters. Thus, discussion about the maximal growth rate seems to be slightly irrelevant here. In the following the authors discuss only one "maximal" growth rate, i.e. only one particular beam velocity.

The statement "The beam pulsed amplifier mechanism is effective for electromagnetic pulses with the duration $tp \leq 10/\gamma_{BP}A \sim 0.1$ s less than the dispersion and nonlinear time scales" needs to be explained in more details.

(2) Author's response

**We would like to thank the Reviewer#2 for the time he/she spent reading, and commenting our manuscript. We have prepared a point-by-point answer to his/her comments below. The responses are marked in bold.**

Reviewer's Comments:

A mechanism responsible for amplification of short VLF electromagnetic pulses based on the cold beam is proposed. A conditional growth rate of short electromagnetic pulses of whistler wave propagating with the wave normal angle equal to the Gendrin angle (presenting the less dispersive wave packet) is calculated. This condition is often discussed as the mechanism of the creating the wave power gap around 0.5 fce. Using the assumption of the cold beam (actually delta function) and making this the key point in the approximation makes the results almost inapplicable to the inner magnetosphere processes. A couple of questions should be addressed before the manuscript might be considered for publication (please, see detailed comments below). Some inaccuracy in the terminology used in the text makes hard to read the manuscript.

The introduction is pure. I think some description of recent results presenting the chorus wave properties, the chorus generation mechanisms (especially beam driven) have to be presented and

discussed. Some discussion of the previous results based on the similar initial conditions (for example, Starodubtsev, et al. (1999) Resonant Cherenkov emission of whistlers by a modulated electron beam. Physics of Plasmas, 6(7), 2862–2869. https://doi.org/10.1063/1.873244 and Pivovarov et al., (2012). Excitation of VLF waves by an electron beam injected into the ionosphere. Journal of Geophysical Research: Space Physics, 100(A9), 17515–7526.https://doi.org/10.1029/95JA01156 and some others from the references therein) have to be included and the new effects obtained in the manuscript need to be emphasized.

**Response:**
**This short theoretical manuscript is addressed to professionals who are well aware of the published achievements of experimental and theoretical research on the chorus radiation in the magnetosphere. Therefore, the bibliography contained only references that are important for this original study.**

Reviewer's Comments:

Condition (5) in the context looks too artificial and needs some more discussion because so cold electron beams are not observed in the magnetosphere that makes all results to be related more to laboratory plasma.

**Response:**
**We derived the condition (5) directly from the Vlasov equation with the stable distribution function $f_\circ(\vec{v})$.**

Reviewer's Comments:

The condition of 'a short electromagnetic pulse' is not well established and explained

**Response:**
**The chosen pulse shape is modeled and corresponds to the shot noise well known in electronics (see, e.g., (Rytov et al., 1989)).**

Reviewer's Comments:

The term 'the Doppler resonances" is not often used for w - kzvz =swB. I suggest to change that to the more conventional ones.

**Response:**
**According to reviewer's suggestion the term "the Doppler resonances" is replaces by "the resonance conditions".**

Reviewer's Comments:

L45: "Cos Theta _BPA=0.6" – please, specify is that definition for the angle of the maximal growth rate? This is not clear here and after. Actually, the authors discuss a very specific case of propagation at the Gendrin angle. This condition gives only one solution (if any) for the particular beam velocity, i.e. properties of the amplified wave are determined by the beam parameters. Thus, discussion about the maximal growth rate seems to be slightly irrelevant here. In the following the authors discuss only one "maximal" growth rate, i.e. only one particular beam velocity.

**Response:**
**It is easy to verify that the function $\gamma(\theta)$ in the equation (15) has maximum at $\cos^2 \theta_{BPA} = 0.6$. We note that, according to the dispersion equation (14), the conditional instability takes place over a wide range of angles with a smaller growth rate.**

Reviewer's Comments:

The statement "The beam pulsed amplifier mechanism is effective for electromagnetic pulses with the duration tp ≤ 10/\gamma_BP A ~ 0.1 s less than the dispersion and nonlinear time scales" needs to be explained in more details.

**Response:**

**The text "less than the dispersion and nonlinear time scales" is replaced by "at the linear stage of the pulse evolution".**

(3) Author's changes in manuscript

Page 4 line 3

The text

"the Doppler resonances"

is replaces by

"the resonance conditions"

Page 7 line 4

The text

"less than the dispersion and nonlinear time scales"

is replaced by

[revised manuscript text omitted]

---

## Author Response (AR2)

**Respond to editor comments**

(1) Comment from Editor

The sentence "Inside the pulse there is a population of electrons which have flown into the interaction region together with the pulse and is fly together with it" (lines 21-22) doesn't read well, especially the "is fly together with it" part. Please revise.

(2) Author's response

**We would like to thank the Editor for important comments. We have prepared a point-by-point answer to comments below. The responses are marked in bold.**

Editor's Comments:

The sentence "Inside the pulse there is a population of electrons which have flown into the interaction region together with the pulse and is fly together with it" (lines 21-22) doesn't read well, especially the "is fly together with it" part. Please revise.

**Response:**

**We revised the sentence and wrote it as follows "Inside a current pulse localization there is a beam of electrons which have entered the interaction region together with the pulse and move jointly with it."**

 (3) Author's changes in manuscript

Page 5 line 21
The text

[revised manuscript text omitted]